# Rustify: Towards Repository-Level C to Safer Rust via Workflow-Guided Multi-Agent Transpiler

## Abstract

Translating C to Rust at the repository level presents unique challenges because of complex dependencies and the differences between C's manual memory model and Rust's strict ownership rules. Existing approaches often overlook repository-wide dependencies and contextual structure, resulting in impractical and potentially unsafe Rust code. We propose **Rustify**, a workflow-guided architecture for repository-level C-to-Rust translation. Rustify decomposes the translation process into modular and role-specific stages, aligned with the structure of source repositories rather than dynamic plan-and-act strategies. These stages address the structural challenges of repository-level C-to-Rust translation, such as determining translation units and resolving cross-file dependencies, with each stage assigned to a dedicated LLM-powered agent. By orchestrating these agents through a structured workflow, Rustify systematically manages repository-wide translation while maintaining semantic equivalence and memory safety. The framework further integrates compiler feedback through tree search–based iterative repair and leverages a dynamic experience base to guide future translations. Experiments show that Rustify consistently produces fully memory-safe Rust code, reducing unsafe code by up to 99% compared to prior approaches. It successfully compiles eight out of nine repositories, raises the test pass rate from 10.5% in the baseline to 86.4%, and achieves high-quality, idiomatic Rust translation with CodeBLEU scores reaching 0.76. A replication repository is available at `https://github.com/rustify712/Rustify`.

## 1 Introduction

C is widely used in system software development for its efficiency and fine-grained memory control Peta (2022) but lacks built-in memory safety features that often lead to issues such as dangling pointers Vintila et al. (2021); Han et al. (2022); Hanley (2023). In contrast, Rust offers compile-time guarantees through its ownership and borrowing mechanisms and enforces memory safety with lifetime annotations while maintaining comparable performance. These features make Rust an increasingly attractive alternative to C Emre et al. (2021). Consequently, there is an urgent demand in industry to migrate existing C-based systems to Rust, particularly in critical domains like operating systems, to enhance overall safety and maintainability.

Existing C-to-Rust translation approaches can be broadly categorized into (a) rule-based and (b) LLM-based approaches. Rule-based approaches Immunant (2022); Shetty et al. (2019); Emre et al. (2021); Han et al. (2022); Ling et al. (2022), such as C2Rust, primarily focus on abstract syntax tree mapping and directly translate C constructs into their Rust equivalents. However, these approaches frequently generate non-idiomatic Rust code that heavily relies on `unsafe` blocks, potentially undermining Rust's core memory safety guarantees. For instance, as illustrated in Appendix A(a), when translating a C program that manipulates a linked list, C2Rust-translated Rust code often relies on raw pointers (*mut Node*) for elements like *head* and *next*, instead of leveraging safe abstractions such as *Box* or *Rc*. This leads to unsafe

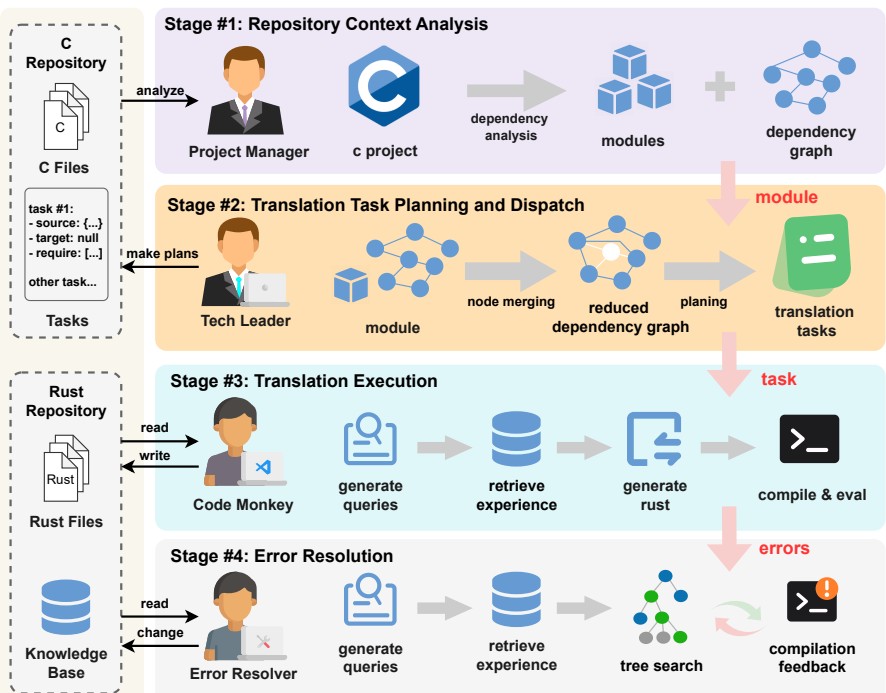

Figure 1: Workflow of Rustify, a multi-agent framework for repository-level C-to-Rust translation. The system includes four specialized LLM agents: `ProjectManager` conducts repository context analysis, `TechLeader` plans and dispatches translation tasks, `CodeMonkey` performs guided translation with experience retrieval, and `ErrorResolver` applies compiler-guided tree search to repair translation errors.

memory operations, with functions like *delete* and *main_0* marked as `unsafe`, indicating that C2Rust fails to fully enforce Rust's safety model.

Recent LLM-based approaches Pan et al. (2024); Eniser et al. (2024); Bhattarai et al. (2024), shown in Appendix A(b), have shown promise in generating more idiomatic and memory-safe Rust code, but they still struggle to meet Rust's stringent compile-time requirements. They frequently encounter issues such as type mismatches, conflicts between mutable and immutable references, and improper ownership transfers. In the LLM-generated code, statements like *\*current = node.next.take();* violate Rust's borrowing rules, as *\*current* is borrowed and reassigned within the same scope, leading to compilation errors. Moreover, while these approaches primarily focus on isolated code snippets, they face significant challenges in repository-level translation—i.e., translating an entire software repository while preserving cross-file dependencies and module relationships. Their limited contextual understanding often results in failures to resolve references to external functions, macros, or type definitions, producing incomplete or incorrect code. This is evident in cases where generated code in *main.rs* reimplements functionality already present in *list.rs*, leading to redundancy and namespace conflicts, as illustrated in Appendix A(b).

Inspired by established software engineering methodologies and development patterns, we propose Rustify, a workflow-guided architecture for repository-level C-to-Rust translation. Unlike dynamic plan-and-act frameworks such as MetaGPT Hong et al. (2023), which depend on runtime planning and sequential execution, Rustify follows a workflow tailored to the structure of source code repositories, ensuring stable collaboration and effective handling of complex dependencies. As depicted in Figure 1, Rustify takes a C repository as input and produces a functionally equivalent, memory-safe Rust repository as output. It systematically manages repository-level dependencies, formulates context-aware translation tasks, performs translation, and iteratively refines the output based on compiler feedback. Each stage is handled by a specialized agent: ProjectManager (stage #1) performs context analysis and module decomposition; TechLeader (stage #2) organizes and performs translation

tasks; CodeMonkey (stage #3) performs translation using chain-of-thought reasoning to preserve semantics; and ErrorResolver (stage #4) applies compiler-guided tree search to repair errors. Additionally, Rustify maintains an evolving knowledge base that captures successful translation and repair experiences, enabling retrieval-augmented reasoning to guide future tasks.

We evaluate Rustify on HumanEval, a private repository, and seven open-source repositories. Experimental results show that Rustify substantially outperforms existing approaches. Compared to C2Rust and Pan et al., it reduces unsafe code by up to 99%,improves the compilation success rate from 15.8% to 100%, and raises test pass rate from 10.5% to 86.4%, while achieving a CodeBLEU score of 0.76. An incremental module analysis further demonstrates the complementary contributions of each component: repository-level planning raises compilation from 15.78% to 57.89%, compiler-guided repair improves it to 68.42% and test pass rate to 57.89%, and experience-based retrieval boosts CodeBLEU from 0.56 to 0.71 while eliminating all unsafe code.

## 2 RUSTIFY

As introduced in Figure 1, Rustify adopts a workflow-guided architecture tailored to repository-level C-to-Rust translation, in contrast to prior dynamic plan-and-act frameworks. This design enables structured coordination among specialized LLM agents for stable and context-aware translation. We now describe each component in detail.

### 2.1 REPOSITORY CONTEXT ANALYSIS

Repository-level translation presents significant challenges due to complex interdependencies among files and functions, including data references, function calls, and global variables, which heavily influence the translation process and outcomes. Existing LLM-based approaches typically rely on concatenating entire files or even complete repositories into a single input prompt. This strategy quickly exceeds the context window limitations of current models and fails to accurately capture inter-file relationships, especially in large and dependency-heavy repositories. As a result, such approaches often struggle to resolve cross-file references and maintain semantic consistency in repository-level translation.

To address these issues, Rustify employs a `ProjectManager` agent to conduct comprehensive repository context analysis. It uses Clang to parse each C source file into AST and records code nodes such as type and function definitions, as summarized in Appendix B. File-level dependencies are then derived by analyzing *#include* directives, while code-level dependencies are obtained from type references and function calls within the AST. Because Clang does not handle conditional compilation directly, Rustify additionally integrates Tree-sitter to extract relevant preprocessor structures. This multi-level analysis yields a dependency graph capturing both file-level and code-level relationships, enabling the `ProjectManager` to partition the repository into independent translation units and assign them to the `TechLeader` for task planning. To support downstream translation, the `ProjectManager` also uses LLMs to summarize each file and the repository as a whole. For example, the dependency graph for the linked list case in Figure 2 determines the translation order of code nodes used in task generation. A detailed description of the `ProjectManager` implementation is provided in Appendix C.

### 2.2 TRANSLATION TASK PLANNING AND DISPATCH

Planning translation at the code node level poses structural challenges in repository-level C-to-Rust migration. Real-world repositories often contain complex dependency graphs spanning functions, types, and macros across multiple files. Without explicit ordering, LLM-based methods may translate nodes independently or in loosely grouped batches, ignoring global dependencies. This lack of coordination can cause duplicated code, missing definitions, and semantic inconsistencies, ultimately affecting correctness and compilation quality.

To address this, the `TechLeader` agent organizes translation tasks in a dependency-aware manner. It uses the dependency graph built during context analysis to perform topological sorting, ensuring referenced definitions are translated first. This preserves correctness and supports parallel execution across independent units.

However, translating each code node individually often results in overly fragmented tasks. Such fine-grained decomposition increases prompt overhead, disperses related logic across tasks, and limits the ability to apply higher-level abstractions in Rust. To address these issues, `TechLeader` further merges translation units using two heuristics:

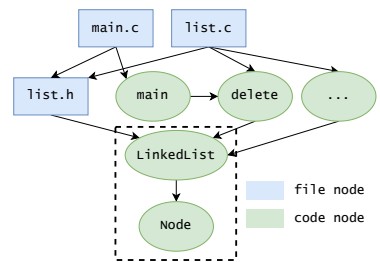

**Dependency-Based Merging.** If two translation units form a tightly coupled pair-one depends solely on the other and vice versa—they are merged into a single task. For instance, as shown in Figure 2, `Node` and `LinkedList` form such a pair and are translated together.

Figure 2: Dependency graph for the linked list, with `LinkedList` and `Node` merged into a single translation unit.

**Line-Based Merging.** For small code elements such as macros or global variables, TechLeader merges them into a combined unit until a minimum size threshold (e.g., 20 lines) is met. This helps reduce overhead and enables more efficient prompting.

The resulting set of merged, dependency-ordered translation tasks is then dispatched to `CodeMonkey` agents. Each agent receives sufficient upstream context to ensure semantic preservation and stability in the generated Rust code. A detailed description of the `TechLeader` implementation is provided in Appendix D.

## 2.3 Experience-Based Knowledge Enhancement

While large language models have shown strong capabilities in general-purpose code generation, they remain limited in repository-level C-to-Rust translation due to insufficient exposure to domain-specific transformation patterns and Rust-specific constraints. In particular, idiomatic usage of Rust constructs, safe memory abstractions, and effective handling of compiler diagnostics are underrepresented in pretraining corpora. As a result, LLMs often produce non-idiomatic code, repeat common translation errors, or fail to resolve compiler feedback consistently across tasks.

To address these limitations, Rustify maintains a dynamic experience knowledge base that augments the reasoning capabilities of LLM agents throughout the workflow. The knowledge base provides reusable transformation strategies and compiler-aware repair solutions collected from previous translation and debugging sessions. It supports both translation and repair phases by offering task-specific guidance and mitigates brittleness in zero-shot prompting.

Rustify organizes accumulated knowledge into two categories:

**Translation experience.** This category captures reusable strategies for translating C constructs into safe and idiomatic Rust code. For example, C raw pointers are systematically translated into abstractions such as `Box<T>`, `Rc<T>`, or `RefCell<T>`, depending on ownership and mutability requirements. Each strategy is annotated with applicability conditions, representative code examples, and known risks, such as runtime borrow violations when overusing `RefCell<T>`.

**Error Resolution experience.** This category consists of error resolution patterns distilled from prior compiler-guided repair sessions. It includes strategies for addressing common issues such as borrow checker violations, lifetime mismatches, and type inference failures. Each entry is indexed by triggering context and typical effects, including scope restructuring, reference cloning, and the use of interior mutability where applicable.

## 2.4 Translation Execution

While repository-level context is essential for translation, it often fails to resolve fine-grained mismatches between C constructs and idiomatic Rust. Large language models struggle in scenarios where training data lacks consistent C-to-Rust patterns or compiler-aware guidance, leading to incorrect ownership handling, inappropriate memory abstractions, and inconsistent translation across related components. To mitigate these issues, Rustify enhances translation with structured experience retrieval and guided decision-making.

Each translation task assigned by the `TechLeader` is processed by the `CodeMonkey` agent via a four-step workflow that integrates context and accumulated translation knowledge to improve stability, correctness, and idiomaticity. For implementation details, see Appendix E.

**Step 1. Experience query generation.** CodeMonkey analyzes the input C code to identify patterns such as raw pointers, manual memory allocation, and shared mutability, and formulates precise queries to the knowledge base (Section 2.3).

**Step 2. Experience retrieval.** The agent retrieves relevant strategies and idiomatic Rust examples, each annotated with applicability conditions and known risks, to guide translation.

**Step 3. Rust code generation.** CodeMonkey generates Rust code using chain-of-thought prompting, informed by context and retrieved knowledge, to preserve semantics and ensure memory safety.

**Step 4. Compilation and evaluation.** Multiple candidates are compiled, and the one with the fewest errors is selected. If none succeed, LLMs rerank the candidates; failing that, the task is escalated to the `ErrorResolver` (Section 2.5).

## 2.5 Error Resolution

While experience-guided translation reduces compile-time errors, semantic violations such as incorrect borrowing or lifetime misuse may still prevent code from compiling. Traditional trial-and-error repair methods often fall into repetitive failure loops, particularly when LLMs are prompted deterministically and generate similar suggestions. To overcome these limitations, Rustify adopts a parallel tree search strategy that explores diverse candidate fixes and selects the most promising one based on compiler feedback.

When translation candidates fail to compile, the `ErrorResolver` agent is invoked to diagnose and resolve the issue. Guided by compiler diagnostics and strategies retrieved from the experience knowledge base (Section 2.3), it performs iterative repair using a greedy tree search to reduce errors while preserving the original intent. A detailed description of the `ErrorResolver` is provided in Appendix F.

**Expansion.** The `ErrorResolver` parses compiler messages to extract error types and locations. It then retrieves relevant fixes, such as for borrow checker violations or type mismatches, and generates diverse repair candidates by varying temperature and prompt structure.

**Evaluation.** Each candidate is compiled, and the output is logged. Candidates are ranked based on compilation success and the number of remaining errors.

**Selection.** The candidate with the fewest errors is selected. If several candidates perform equally well, the one closest to the original translation is preferred. Successful repairs are merged into the repository, while unresolved cases proceed through further iterations until completion or termination.

## 3 Experiments

We conduct experiments to evaluate the effectiveness of our approach in translating C code to safe and idiomatic Rust.

Table 1: Statistics of C Programs. Level: scope of translation (Code/File/Repository level), Loc: lines of C code, # Files: number of source files, Avg Loc: average lines of code per file, # Tasks: number of translation tasks, # Test Cases: number of test cases for verification

| Dataset | Level | Loc | # Files | Avg Loc | # Tasks | # Test Cases |
|---------|-------|-----|---------|---------|---------|--------------|
| HumanEval | Code | 4408 | 164 | 26 | 164 | 164 |
| urlparser | File | 564 | 1 | 564 | 22 | 1 |
| libtree | File | 1837 | 1 | 1837 | 103 | 9 |
| json.h | File | 3440 | 1 | 3440 | 71 | 44 |
| quadtree | Repository | 407 | 5 | 81 | 27 | 4 |
| buffer | Repository | 453 | 2 | 226 | 25 | 17 |
| rgba | Repository | 463 | 2 | 231 | 18 | 5 |
| binn | Repository | 4396 | 2 | 2198 | 383 | 10 |
| private | Repository | 9499 | 38 | 249 | 274 | 125 |

## 3.1 DATASETS

To evaluate the performance of our approach, we use a diverse set of datasets spanning code-level, file-level, and repository-level translation. Our evaluation includes the HumanEval benchmark Chen et al. (2021), private data structure benchmark, and seven real-world open-source repositories Zhang et al. (2023). These datasets include test cases crafted by expert teams, as well as the translated Rust reference code. Table 1 provides a detailed overview of these datasets.

## 3.2 BASELINE

We compare Rustify with both traditional rule-based and recent LLM-based approaches for C-to-Rust translation. For rule-based approaches, we evaluate C2Rust Immunant (2022), a widely used transpiler that performs direct AST mapping from C to Rust. For LLM-based approaches, we implement the method proposed by Pan et al. Pan et al. (2024), which leverages LLMs for code translation and iteratively refines the output using compiler feedback. We also include MetaGPT Hong et al. (2023), a multi-agent framework that applies task planning and role-based collaboration to software development tasks, including translation. Additionally, LLMs such as DeepSeek-v3, GPT-4, and Claude-3.5-Sonnet are employed as agents within Rustify for comparison.

## 3.3 EXPERIMENTAL SETUP

We evaluate translation performance using four metrics: **Unsafe Ratio**, **Comp Rate**, **Pass Rate**, and **CodeBLEU**. Unsafe Ratio measures the proportion of lines in the generated Rust code marked as `unsafe`, reflecting memory safety. Comp Rate measures the percentage of files that compile successfully, and Pass Rate quantifies the proportion of test cases passed. CodeBLEU evaluates translation quality in terms of syntactic and semantic similarity to reference code. Detailed metric formulations and retrieval configurations are provided in Appendix G.

## 3.4 EVALUATION OF RUSTIFY

We evaluate Rustify's effectiveness in translating C code to Rust, focusing on code safety, correctness, functionality, and overall quality. The results are reported in Table 2

**Safety Evaluation.** The Unsafe Ratio shows that C2Rust generates unsafe code in most cases, ranging from 26.5% to 99%. Pan et al. reduces this for small tasks but still uses unsafe code in complex repositories. MetaGPT improves safety on several benchmarks but remains unstable overall. Rustify consistently achieves a 0% Unsafe Ratio across all datasets and models, reducing unsafe code by up to 99% and surpassing both baselines.

Table 2: Experimental results of C2Rust, the approach proposed by Pan et al., MetaGPT and Rustify on DeepSeek, GPT-4o, and Claude-3.5-Sonnet. ✓ indicates 100% success and × denotes execution failure.

| Metric | Approach | | HumanEval | urlparser | libtree | json.h | quadtree | buffer | rgba | binn | private |
|---|---|---|---|---|---|---|---|---|---|---|---|
| Unsafe Ratio | C2Rust | - | 84.87% | 84.13% | 86.41% | 96.05% | 76.21& | 79.40% | 26.54% | 99% | 93.58% |
| | Pan et al. | DeepSeek-v3 | 0% | 0 | 16.71% | 25.36% | 0.56% | 30.85% | 0 | 49.86% | 19.02% |
| | | GPT-4o | 0 | 0 | 0 | 16.33% | 0 | 0 | 0 | 12.11% | 13.61 % |
| | | Claude-3.5-Sonnet | 0 | 0 | 6.3% | 9.18% | 0 | 0 | 0 | 11.36% | 7.35% |
| | MetaGPT | DeepSeek-v3 | 0% | 0% | 5.65% | 34.93% | 1.66% | 1.81% | 0% | 52.33% | 11.47% |
| | | GPT-4o | 0% | 0% | 5.61% | 0% | 0% | 0% | 0% | 11.58% | 16.11% |
| | | Claude-3.5-Sonnet | 0% | 0% | 4.39% | 0% | 0% | 0% | 0% | 8.76% | 9.45% |
| | Rustify | DeepSeek-v3 | 0 | 0 | 0 | 0 | 0 | 0 | 0 | 0 | 0 |
| | | GPT-4o | 0 | 0 | 0 | 0 | 0 | 0 | 0 | 0 | 0 |
| | | Claude-3.5-Sonnet | 0 | 0 | 0 | 0 | 0 | 0 | 0 | 0 | 0 |
| Comp Rate | C2Rust | - | ✓ | ✓ | × | ✓ | ✓ | ✓ | ✓ | ✓ | ✓ |
| | Pan et al. | DeepSeek-v3 | ✓ | ✓ | × | × | 50% | ✓ | ✓ | × | 68.42% |
| | | GPT-4o | ✓ | ✓ | × | × | 75% | ✓ | × | × | 57.89% |
| | | Claude-3.5-Sonnet | ✓ | ✓ | × | × | 75% | ✓ | × | × | 73.68% |
| | MetaGPT. | DeepSeek-v3 | 95.73% | ✓ | 20% | × | 16.67% | ✓ | × | 28.57% | 84.21% |
| | | GPT-4o | 94.94% | 50% | × | × | 60% | ✓ | ✓ | 44.44% | 84.21% |
| | | Claude-3.5-Sonnet | ✓ | ✓ | 7.14% | × | 33.33% | ✓ | ✓ | 50% | 89.47% |
| | Rustify | DeepSeek-v3 | ✓ | ✓ | ✓ | ✓ | ✓ | ✓ | ✓ | × | 89.47% |
| | | GPT-4o | ✓ | ✓ | ✓ | × | ✓ | ✓ | ✓ | × | 73.68% |
| | | Claude-3.5-Sonnet | ✓ | ✓ | ✓ | ✓ | ✓ | ✓ | ✓ | × | ✓ |
| Pass Rate | C2Rust | × | ✓ | ✓ | × | ✓ | ✓ | ✓ | ✓ | ✓ | ✓ |
| | Pan et al. | DeepSeek-v3 | 94.51% | × | × | × | × | 88.23% | 40% | - | 29.93% |
| | | GPT-4o | 93.29% | × | × | × | × | 70.58% | × | × | 22.4% |
| | | Claude-3.5-Sonnet | 95.12% | × | × | × | 50% | ✓ | ✓ | × | 37.37% |
| | MetaGPT. | DeepSeek-v3 | 91.46% | × | × | × | × | ✓ | × | × | 22.4% |
| | | GPT-4o | 90.85% | × | × | × | 50% | 88.23% | 60% | × | 32.8% |
| | | Claude-3.5-Sonnet | 99.39% | × | × | × | × | ✓ | × | × | 82.4% |
| | Rustify | DeepSeek-v3 | ✓ | ✓ | 33% | 68% | ✓ | ✓ | ✓ | × | 80% |
| | | GPT-4o | ✓ | ✓ | 22% | 48% | ✓ | ✓ | ✓ | × | 74.4% |
| | | Claude-3.5-Sonnet | ✓ | ✓ | 33% | 63% | ✓ | ✓ | ✓ | × | 86.4% |
| CodeBLEU | C2Rust | - | 0.29 | 0.6 | 0.17 | 0.16 | 0.32 | 0.25 | 0.22 | 0.29 | 0.35 |
| | Pan et al. | DeepSeek-v3 | 0.36 | 0.72 | 0.38 | 0.19 | 0.59 | 0.41 | 0.29 | 0.25 | 0.42 |
| | | GPT-4o | 0.35 | 0.73 | 0.38 | 0.31 | 0.54 | 0.41 | 0.43 | 0.33 | 0.45 |
| | | Claude-3.5-Sonnet | 0.36 | 0.4 | 0.4 | 0.32 | 0.55 | 0.32 | 0.47 | 0.34 | 0.47 |
| | MetaGPT. | DeepSeek-v3 | 0.27 | 0.70 | 0.39 | 0.17 | 0.63 | 0.52 | 0.42 | 0.43 | 0.50 |
| | | GPT-4o | 0.33 | 0.69 | 0.25 | 0.11 | 0.57 | 0.34 | 0.33 | 0.19 | 0.43 |
| | | Claude-3.5-Sonnet | 0.41 | 0.71 | 0.51 | 0.40 | 0.57 | 0.50 | 0.44 | 0.40 | 0.69 |
| | Rustify | DeepSeek-v3 | 0.42 | **0.76** | 0.47 | 0.41 | **0.69** | 0.69 | 0.53 | 0.47 | 0.71 |
| | | GPT-4o | 0.4 | 0.75 | 0.45 | 0.42 | 0.66 | 0.66 | **0.55** | 0.56 | 0.7 |
| | | Claude-3.5-Sonnet | **0.42** | 0.75 | **0.47** | **0.42** | 0.68 | **0.69** | 0.54 | **0.56** | **0.71** |

**Correctcness Evaluation.** C2Rust compiles most datasets except libtree, where it fails due to macros. Pan et al. often fails on large or dependency-heavy repositories. MetaGPT improves compilation in some cases, especially with Claude, but cannot handle all complex dependencies. Rustify compiles eight out of nine repositories, thanks to its context-aware planning and error repair.

**Functionality Evaluation.** Pass Rate measures whether generated Rust code passes tests. C2Rust performs well except on libtree. Pan et al. handles small tasks but struggles with larger ones. MetaGPT improves in some cases, with Claude achieving 82.4% on private, but results vary. Rustify performs more reliably, reaching 86.4% on private and showing strong functional preservation overall.

**Quality Evaluation.** CodeBLEU evaluates idiomaticity. C2Rust scores low, often producing C-like Rust. Pan et al. performs better but inconsistently. MetaGPT achieves similar improvements but lacks stability on large projects. Rustify achieves the highest and most stable scores, up to 0.76, and consistently generates idiomatic and semantically correct Rust code.

**Discussion.** The results confirm that large language models possess a certain level of understanding of C and Rust, enabling them to produce partially correct and idiomatic translations. However, their limited global awareness makes them unreliable for repository-level translation. For example, dynamic planning systems like MetaGPT generate task sequences based on local file inspection, but often fail to maintain correct translation order, leading to duplicated definitions, missing references, and semantic inconsistencies. These failures are especially common in large files, where only partial context is available during planning. Moreover, the lack of structural grounding causes these models to hallucinate logic, resulting in incomplete or misaligned translations.

Rustify addresses these limitations through a workflow explicitly aligned with the structural characteristics of source repositories. Instead of relying on emergent planning, it defines four concrete stages: context analysis, task planning, translation, and error repair. Each stage is handled by a dedicated LLM agent. This separation ensures that dependencies are preserved, translations are modular and context-aware, and errors are resolved iteratively using compiler feedback and accumulated experience. These components work together to deliver substantial improvements in compilation success, safety, and semantic fidelity across all evaluated benchmarks.

Table 3: Incremental module evaluation results: starting from Base (which includes only the translation module), then incrementally adding context analysis, repair, and knowledge modules.

| Configuration | Metric | HumanEval | urlparser | libtree | json.h | quadtree | buffer | rgba | binn | private |
|---|---|---|---|---|---|---|---|---|---|---|
| Base | Unsafe Rate | 0 | 0 | 0 | 0 | 0 | 0 | 0 | 0 | 0 |
| | Comp Rate | 74.39% | × | × | × | 50% | × | × | × | 15.78% |
| | Pass Rate | 72.56% | × | × | × | 50% | × | × | × | 10.52% |
| | CodeBLEU | 0.41 | 0.70 | 0.38 | 0.35 | 0.60 | 0.51 | 0.47 | 0.34 | 0.56 |
| + Context | Unsafe Rate | 0 | 0 | 0 | 7.51% | 0 | 0 | 0 | 9.44% | 3.41% |
| | Comp Rate | ✓ | ✓ | × | × | 50% | ✓ | ✓ | × | 57.89% |
| | Pass Rate | ✓ | × | × | × | 50% | 88.23% | 40% | × | 15.78% |
| | CodeBLEU | 0.38 | 0.72 | 0.41 | 0.35 | 0.55 | 0.59 | 0.47 | 0.35 | 0.47 |
| + Repair | Unsafe Rate | 0 | 0 | 0 | 6.98% | 0 | 0 | 0 | 9.17% | 5.49% |
| | Comp Rate | ✓ | ✓ | × | × | ✓ | ✓ | ✓ | × | 68.42% |
| | Pass Rate | ✓ | ✓ | × | × | ✓ | ✓ | ✓ | × | 57.89% |
| | CodeBLEU | 0.41 | 0.74 | 0.46 | 0.39 | 0.66 | 0.64 | 0.54 | 0.52 | 0.69 |
| + Knowledge | Unsafe Rate | 0 | 0 | 0 | 0 | 0 | 0 | 0 | 0 | 0 |
| | Comp Rate | ✓ | ✓ | ✓ | ✓ | ✓ | ✓ | ✓ | × | ✓ |
| | Pass Rate | ✓ | ✓ | 33% | 63% | ✓ | ✓ | ✓ | - | 86.4% |
| | CodeBLEU | 0.42 | 0.75 | 0.47 | 0.42 | 0.68 | 0.69 | 0.54 | 0.56 | 0.71 |

## 3.5 Incremental Module Evaluation

To better characterize the contribution of each module in our workflow-guided framework, we conduct an incremental module evaluation using Claude-3.5-Sonnet, as shown in Table 3. We start with the basic configuration containing only the translation module. Then, we incrementally add the context analysis module for repository-level planning, the repair module for compiler-guided tree search, and finally the knowledge module for experience-based retrieval and reuse.

**Base.** The base configuration performs translation without compiler feedback or repair. It achieves moderate CodeBLEU scores, such as 0.56 on the private repository and 0.70 on urlparser, but lacks functional robustness. On the private repository, only 15.78% of files compile successfully, and just 10.52% of test cases pass. Without any mechanism for error correction, most real-world translation tasks fail to complete reliably.

**+ Context.** Introducing the context module enables dependency-aware planning across the repository. This improves structural alignment, raising the compilation rate on the private repository to 57.89% and the pass rate to 15.78%. However, some unsafe code is introduced (e.g., 7.51% in json.h), and the CodeBLEU score on several datasets slightly declines, suggesting that structural correctness alone does not guarantee semantic or idiomatic consistency.

**+ Repair.** The repair module applies compiler-guided tree search to correct translation errors. This leads to marked improvements in both correctness and safety. On the private repository, compilation success reaches 68.42%, test pass rate improves to 57.89%, and the unsafe rate drops significantly. CodeBLEU also improves, increasing from 0.47 to 0.69, reflecting better semantic alignment and stability.

**+ Knowledge.** The knowledge module draws on prior translation and repair experience to further enhance performance. On the private repository, it achieves full compilation, raises the pass rate to 86.4%, and eliminates all unsafe code. The CodeBLEU score also rises to 0.71, indicating improved idiomaticity and consistency across translation tasks.

**Discussion.** The incremental evaluation confirms that each module in Rustify addresses a specific limitation in repository-level C-to-Rust translation. The context module improves structural consistency through dependency-aware planning, the repair module enhances correctness and safety by incorporating compiler feedback, and the knowledge module boosts translation stability and idiomatic quality via reuse of past strategies. Notably, the introduction of the repair and knowledge modules yields the most significant improvements: for instance, on the private repository, compilation increases from 15.78% to 100%, and test pass rate rises from 10.52% to 86.4%. These results demonstrate that structural planning alone is insufficient, and that compiler-guided correction and experience reuse are crucial for producing reliable and idiomatic Rust code.

Nevertheless, Rustify still falls short of full test coverage in certain repositories. Most failures stem from complex pointer manipulations, intricate control logic, or project-specific patterns in the original C code that are difficult for the compiler to diagnose and are not yet well represented in the experience base. This highlights the need for future extensions such as semantic test oracles, runtime-guided refinement, or broader experience mining to further improve translation completeness and correctness.

## 4 RELATED WORK

**C-to-Rust Translation** Recent work on translating C code to Rust has explored both rule-based and LLM-based approaches. Rule-based methods, such as `C2Rust`, demonstrate the feasibility of automated migration but often generate Rust code that heavily relies on `unsafe` blocks, thus demanding extensive manual refinement to achieve idiomatic Rust. Efforts to improve rule sets have targeted more complex transformations, including smart pointer conversions and synchronization primitives Emre et al. (2023); Hong & Ryu (2023; 2024); Emre et al. (2021), yet the core challenge of balancing high-level automation with Rust's stringent memory safety remains. On the other hand, the emergence of lLLMs has opened new directions for C-to-Rust translation. Early experiments combine LLM-based code generation with formal verification and fuzzing-based testing to validate correctness and safety Eniser et al. (2024); Hong & Ryu (2023); Pan et al. (2024). Despite promising initial results, these approaches face significant scalability issues due to model context window constraints and the inherent difficulty of preserving semantic consistency across large repositories.

**Multi-Agent Collaboration** Recent research on LLM-based multi-agent systems has introduced innovative methodologies for collaborative task-solving and simulation. The framework AutoGen Wu et al. (2023) enables dynamic multi-agent coordination through decentralized conversational interactions, supporting complex task execution across diverse domains. MetaGPT Hong et al. (2023) embeds standardized human workflows into agent roles, structuring collaboration in software development by assigning specialized responsibilities such as product management and coding. Meanwhile, self-collaboration approaches Dong et al. (2024) leverage iterative communication among specialized agents to autonomously refine code generation, balancing individual expertise with collective reasoning. These works highlight advancements in agent profiling, communication mechanisms, and adaptive capability acquisition for multi-agent systems.

## 5 CONCLUSION

We propose Rustify, a workflow-guided architecture for repository-level C-to-Rust translation that divides the process into four LLM-driven stages: context analysis, task planning, translation, and error repair. This structured approach enables stable and context-aware translation while preserving memory safety. Experiments show that Rustify significantly outperforms both rule-based and LLM-based baselines. It eliminates unsafe code in all tested cases, successfully compiles 8 out of 9 repositories, and achieves a test pass rate of 86.4%. Incremental evaluation highlights the importance of each module in improving correctness, safety, and idiomaticity. Future work will aim to further enhance semantic accuracy and functional completeness across more diverse repositories.

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

## A    Translation Challenges

To motivate the design of Rustify, we provide a comparative example highlighting common failure modes in rule-based and LLM-based C-to-Rust translation methods.

```rust
// list.rs
// ... other code
#[derive(Copy, Clone)]
#[repr(C)]
pub struct Node {
    pub value: libc::c_int,
    pub next: *mut Node,                raw pointer
}
#[derive(Copy, Clone)]
#[repr(C)]
pub struct LinkedList {
    pub head: *mut Node,
    pub length: libc::c_int,
}
// ... other functions
#[no_mangle]
pub unsafe extern "C" fn delete(
    mut list: *mut LinkedList, mut value: libc::c_int
) {
    if list.is_null() || ((*list).head).is_null() {
        return;
    }
    let mut current: *mut Node = (*list).head;
    let mut previous: *mut Node = 0 as *mut Node;
    while !current.is_null() {
        if (*current).value == value {
            if previous.is_null() {
                (*list).head = (*current).next;
            } else {
                (*previous).next = (*current).next;
            }
            free(current as *mut libc::c_void);
            (*list).length -= 1;
            (*list).length;
            return;
        }
        previous = current;
        current = (*current).next;
    }
}

// main.rs                                unsafe block
// ... other code
unsafe fn main_0() -> libc::c_int {
    // ... create a new list and insert some elements
    delete(list, 2 as libc::c_int);
    return 0 as libc::c_int;
}
pub fn main() {
    unsafe { ::std::process::exit(main_0() as i32) }
}
```

**(a) C2Rust Result**

```rust
// list.rs
pub struct Node {
    value: i32,
    next: Option<Box<Node>>,
}
pub struct LinkedList {
    head: Option<Box<Node>>,
    length: i32,
}
impl LinkedList {
    // ... other functions
    pub fn delete(&mut self, value: i32) {
        let mut current = &mut self.head;
        while let Some(ref mut node) = current {
            if node.value == value {
                *current = node.next.take();   // error
                self.length -= 1;
                return;                compilation error
            }
            current = &mut node.next;   // error
        }
    }
}

// main.rs    redefinition instead of importing from list.rs
mod list {
    struct Node {
        value: i32,
        next: Option<Box<Node>>,
    }
    pub struct LinkedList {
        head: Option<Box<Node>>,
        length: usize,
    }
    impl LinkedList {
        // other functions
        pub fn delete(&mut self, value: i32) {
            let mut current = &mut self.head;
            while let Some(ref mut node) = current {
                if node.value == value {
                    *current = node.next.take();
                    self.length -= 1;
                    return;
                }
                current = &mut node.next;
            }
        }
    }
}
fn main() {
    // create a new list and insert some elements
    list.delete(2);
}
```

**(b) LLM Result**

Figure 3: Challenges in C-to-Rust translation. The figure compares (a) a rule-based approach (C2Rust) and (b) an LLM-based approach on a linked list example.

## B    Code Node Categories

To enable fine-grained translation and dependency tracking, Rustify extracts and categorizes semantic elements from C code into distinct node types. This section enumerates the code node categories used during repository-level context analysis.

Table 4: Code node categories extracted during context analysis.

| Category | Description |
| --- | --- |
| Macro | Preprocessor directive for text substitution. |
| Typedef | A type definition using `typedef x y`. |
| Struct | A data structure declared using `struct` or `typedef struct x {...}` y. |
| Union | A data structure declared using `union` or `typedef union x {...}` y. |
| Enum | A data structure declared using `enum` or `typedef enum x {...}` y. |
| Function | C function or inline function. |
| Variable | Global variable. |

## C    ProjectManager

The `ProjectManager` agent conducts repository-level context analysis. It parses C source files into abstract syntax trees, constructs dependency graphs, summarizes file-level functionality, and partitions the repository into modular translation units. This section presents its execution workflow and prompt design.

---

**Algorithm 1** ProjectManager Lifecycle Overview

---

 1: **Step 1: Load the C project and perform initialization**
 2:     Identify all source files and register the project in internal state
 3: **Step 2: Generate summaries for each source file**
 4:     Use LLM to extract and store the purpose of each C/C++ file
 5: **Step 3: Analyze file dependencies**
 6:     Construct a dependency graph and group related files into logical modules
 7: **Step 4: Create the target Rust project**
 8:     Initialize module metadata using previously summarized information
 9: **Step 5: Assign translation tasks to Tech Leaders**
10:     Dispatch modules for translation, manage concurrency, and track progress
11: **End:** All modules successfully translated and assembled

---

**File Summary Prompt for Translation Preparation**

**Task:** Before translating a C project to Rust, read and fully understand the C source files. Write a summary for each file, outlining its main content, functionality, and purpose. The goal is to help developers understand the overall architecture and responsibilities of each part, so module boundaries can be defined more effectively during the translation process.
**Context:**
**Project Structure:** {project_structure}
**File to Summarize:** {filepath}
**File Content:** {content}
**Instructions:**

- Use clear and simple language.

- Avoid technical jargon or low-level implementation details.

- Summarize the file's main responsibilities, purpose, and how it interacts with other parts of the project.
- Keep the summary concise — no more than 300 words.

**Example Summary:**
The file xxx mainly handles user input and data validation. It performs basic checks, format conversions, and error reporting to ensure data correctness. The processed input is passed to downstream logic. This module plays a key role in safeguarding external inputs, offering reliable data for the rest of the system.

## D TECHLEADER

The `TechLeader` agent orchestrates translation planning. It schedules code node translation based on topological order, merges related units to reduce fragmentation, and generates task summaries to guide downstream agents. This section describes its operational steps and prompting templates.

---
**Algorithm 2** TechLeader Lifecycle Overview
---
1: **Step 1: Initialize the module project**
2:   Create the Rust project based on the module's description and related source files
3: **Step 2: Generate explanations for translation tasks**
4:   Use LLM to summarize the purpose of each task to assist downstream agents
5: **Step 3: Dispatch translation tasks to CodeMonkeys**
6:   Schedule tasks according to dependency order and manage concurrent execution
7: **Step 4: Wait for task completion**
8:   Monitor all tasks until completed and mark the module as done
9: **End:** Module translation finished and results stored
---

**Summary Prompt for TechLeader**

**Task:** Read and analyze a C code snippet, then produce a concise, plain-language description of its main functionality, purpose, and role.
**Context:**
**File Summary:** {file_summary}
**Code Snippet:** {code_nodes}
**Instructions:**

- Write a single coherent paragraph (no more than 150 words) describing the code's main functionality, inputs, outputs, and its role in the file.
- Emphasize the code's role within the file and how it interacts with other modules or components.

# E CodeMonkey

CodeMonkey performs translation by integrating retrieved experience with repository context. It formulates experience queries, generates idiomatic Rust candidates via chain-of-thought prompting, and selects the most compilable result. This section includes its full workflow and associated prompts.

---

**Algorithm 3** CodeMonkey Lifecycle Overview

---

1: **Step 1: Generate experience queries**
2:  Analyze the C code to identify constructs like raw pointers or manual memory handling, then formulate experience-based queries to guide translation.
3: **Step 2: Retrieve relevant translation strategies**
4:  Query the experience base and retrieve idiomatic Rust patterns with safety annotations and applicability conditions.
5: **Step 3: Generate Rust translation candidates**
6:  Use retrieved experience and translation context to produce Rust code via chain-of-thought prompting, optionally generating multiple candidates with varied sampling temperatures.
7: **Step 4: Compile and evaluate candidates**
8:  Compile each candidate using `cargo check`, and select the best one based on compiler success and minimal error count.
9: **Step 5: Record translation experience**
10:  If a successful translation or fix is achieved, extract reusable patterns and store them in the experience base; otherwise, mark the task as `unimplemented`.
11: **End:** Task completes when compilation succeeds or errors are resolved by the **ErrorResolver**.

---

**Experience Query Prompt for CodeMonkey**

**Task:** Before translating the given C/C++ code, analyze it thoroughly and identify critical features that may require special attention in Rust. Based on your analysis, write 1–3 precise and actionable experience-based queries.
**Context:**
**File Summary:** {file_summary}
**C/C++ Code Snippet:** {code_snippet}
**Focus Areas:**

- **Macros and Preprocessor Directives:** C macros have no direct Rust equivalent; alternatives may include constants or functions.

- **Memory Management:** C uses raw pointers and manual memory control, whereas Rust relies on ownership and borrowing.

- **Ownership and Borrowing:** Critical for safe aliasing and memory access patterns in Rust.

- . . .

**Instructions:**

- Generate 1–3 focused, concrete questions that relate directly to the code snippet.

- Avoid trivial or generic questions (e.g., "Does Rust have a macro system?").

- Ensure each query addresses a transformation-relevant challenge or ambiguity.

**Examples:** . . . (few-shot examples omitted for brevity)

---

**Translation Prompt for CodeMonkey**

**Task:** Read and understand the C/C++ code, consider whether there is an equivalent implementation in Rust, and translate it into Rust code.
**Context:**

**Project Description:** {project description}
**Project Structure:** {project structure}
**Related File Contents:** {related file contents}
**Translation Task:** {current translation task}
**Translation Rules:**

- Use `Option` and `Result` to gracefully handle potential errors or exceptional cases.

- Use references or smart pointers, avoid raw pointers and ensure memory safety.

- . . .

**Experiences:**

- **How to safely implement dynamically sized array allocation in Rust?**
  In Rust, dynamic arrays can be safely allocated using either `Vec<T>` or `Box<[T]>`.

- . . .

## F    ERRORRESOLVER

The `ErrorResolver` agent applies compiler-guided repair using tree search. It diagnoses compiler errors, retrieves resolution strategies from prior experience, explores fix candidates, and selects the minimal, successful patch. This section details its repair process and corresponding prompt structure.

---

**Algorithm 4** ErrorResolver Lifecycle Overview

---

1: **Step 1: Generate targeted repair queries**
2:    Analyze compiler diagnostics and code context to formulate precise repair questions
3: **Step 2: Retrieve relevant fix strategies**
4:    Query the translation memory or external sources for idiomatic Rust fixes and known patterns
5: **Step 3: Explore candidate fixes via MCTS**
6:    Use Monte Carlo Tree Search to explore and prioritize diverse fix candidates under guidance of retrieved strategies
7: **Step 4: Apply and verify patch candidates**
8:    Apply the selected fix, recompile, and log results; score outcomes based on error reduction and semantic plausibility
9: **Step 5: Iterate until fixed or exhausted**
10:    Repeat exploration and application loop until compilation succeeds or retry limit is reached
11: **End:** If successful, finalize fix and update memory; otherwise, fallback to `unimplemented` output

---

**Repair Prompt for ErrorResolver**

Here is the current Rust code of the file:
{code}
The compiler errors are as follows:
{errors}
Here are the expert insights summarized during error fixing:
{experience}
You need to perform deep reasoning and analysis to identify the root causes of the errors and fix them so that the code can compile successfully. First, analyze the description of each error to understand what type of issue it represents. Next, consider the common causes of such errors and how they are typically resolved — you may refer to expert insights as needed. Finally, propose potential solutions for each error and attempt to fix them one by one, providing the corrected code for each.

## G   EVALUATION METRICS AND EXPERIMENT CONFIGURATION

**Metric Formulations.**   We adopt four standard metrics to evaluate translation performance:

- **Unsafe Ratio:** $\frac{N_{\text{unsafe}}}{N_{\text{total}}}$, where $N_{\text{unsafe}}$ is the number of lines marked as `unsafe` in the translated Rust code, and $N_{\text{total}}$ is the total number of lines. This metric quantifies the extent of memory-unsafe code generated.

- **Comp Rate:** $\frac{F_{\text{comp}}}{F_{\text{total}}}$, where $F_{\text{comp}}$ is the number of translation units that compile successfully, and $F_{\text{total}}$ is the total number of translated units. It reflects syntactic and type-level correctness.

- **Pass Rate:** $\frac{T_{\text{pass}}}{T_{\text{total}}}$, measuring the proportion of passed translation tests relative to total test cases. It evaluates functional correctness with respect to expected behavior.

- **CodeBLEU:** A composite metric combining lexical similarity, syntax structure, and semantic preservation. It is computed as:

$$\text{CodeBLEU} = \alpha B_{\text{ngram}} + \beta B_{\text{weight}} + \gamma B_{\text{syn}} + \delta B_{\text{sem}},$$

  with weights $\alpha = 0.1$, $\beta = 0.1$, $\gamma = 0.4$, and $\delta = 0.4$.

**Retrieval Configuration.**   During experience retrieval (Section 2.3), we apply a hybrid search strategy:

- **Dense retrieval:** Qwen text-embedding-v3 is used to encode translation tasks and experience entries into vector representations, enabling similarity-based retrieval.

- **Sparse retrieval:** BM25 is applied in parallel to capture lexical overlap, particularly useful for matching common error tokens or syntax-specific patterns.

