# OpenReview forum: "Rustify: Towards Repository-Level C to Safer Rust via Workflow-Guided Multi-Agent Transpiler"
_ICLR.cc/2026/Conference — ICLR 2026 Conference Withdrawn Submission_

### Official Review · Reviewer_ScML · 2025-10-25

**Soundness:** 2
**Presentation:** 2
**Contribution:** 2
**Rating:** 2
**Confidence:** 4

**Summary:**

This paper presents Rustify, a workflow-guided, multi-agent framework for repository-level C-to-Rust translation, comprising four components: ProjectManager, TechLeader, CodeMonkey, and ErrorResolver. The results show that Rustify successfully compiles eight out of nine repositories.

**Strengths:**

- A workflow-guided, multi-agent architecture is proposed that decomposes the translation process into four specialized stages.

- A dynamic experience base and compiler-guided tree search are introduced for iterative repair.

- Experiments against multiple baselines show that the approach outperforms existing methods.

**Weaknesses:**

- There is limited empirical support for the repository-level claims. The authors report that Rustify successfully translates eight out of nine repositories. However, these include one code-level benchmark (HumanEval) and four small projects (<600 LOC), which do not demonstrate effectiveness on large codebases with complex cross-file dependencies.

- The paper should provide a systematic study of common C → Rust translation errors at the repository level, including which error types are most frequent, which are successfully repaired, and which remain challenging.

**Questions:**

- No data are provided on translation speed, token usage, or API calls, making it difficult to assess the practicality and scalability of this computationally intensive method.

- The paper does not discuss whether the framework can be generalized to other programming language translation tasks, and evaluation on larger, more diverse repository datasets such as RepoTransBench is needed.

- Workfolw is linear and unidirectional.

- The dynamic experience base functions as a local memory, but its technical details and scope (project-specific or shared) are unclear.

---

### Official Review · Reviewer_UR2t · 2025-10-28

**Soundness:** 2
**Presentation:** 3
**Contribution:** 2
**Rating:** 2
**Confidence:** 4

**Summary:**

This paper proposes Rustify, a workflow-guided architecture for the repository-level C-to-Rust translation task. The approach decomposes the translation task into modular with different role-specific agents. The experiments show that the proposed approach Rustify outperforms baselines in generating more safer, more compilation correct, and more functionality correct Rust code.

**Strengths:**

The paper targets an important problem, i.e., repository-level C-to-Rust translation.

The paper is well organized.

**Weaknesses:**

The novelty of the approach is limited. In particular, the workflow-guided architecture has been widely explored in existing agents for software development work[d,e,f,g]. The sperate roles of project managers and code Monkey (i.e., programmers) are not new ideas; in addition, using the error message as feedback has also been widely adopted in many code generation and comprehension tasks, even for code translation task[b,c]; in addition, decomposing the repository to modules have also been explored in existing repository-level code translation (i.e., Alphatrans[a]). Therefore, the major novelty of this work remains questionable compared to existing literature. More justification is definitely necessary to address this limitation.

[a] AlphaTrans: A Neuro-Symbolic Compositional Approach for Repository-Level Code Translation and Validation. FSE’25

[b] Lost in translation: A study of bugs introduced by large language models while translating code. ICSE’24

[c] Exploring and Unleashing the Power of Large Language Models in Automated Code Translation. FSE’24

[d] Experimenting a new programming practice with llms

[e] When llm-based code generation meets the software development process

[f] Communicative agents for software development

[g] Multi-agent software development through cross-team collaboration


The evaluation is insufficient, in terms of projects, metrics, and baselines.

Evaluation projects. The criteria of projects selection is unclear, especially some project (e.g,. HumanEval) seems to be a very unreasonable project for the repository-level C-to-Rust translation task evaluation. In particular, HumanEval is a code generation tasks, while each of the coding task is a separate problem without any dependency. It is very unsuitable to use the benchmark as a repository-level translation task, which is different from real-world repository-level translation task (as the challenge here is the dependencies among code files and functions.) It further comes to the question how authors select the evalution projects and whether they are truly suitable projects for this task.

In addition, the test sufficiency of studied projects remains unclear. As shown in Table-1, the number of tests is very limited for one repository. For example, only one test for urlparser projects, and only 10 tests for the binn projects (with over 4k lines of code). It is necessary to present the sufficiency of the tests (e.g., the line and branch coverage). Otherwise, the calculated metrics (i.e., pass rate) is not convincing, which might overlook many potential semantic errors.

More recent and state-of-the-art code translation techniques are missed as baselines. In particular, [a,b,c,d] are more recent LLM-driven code translation techniques, that are very relevant with the current approach. It is necessary to compare with them or give the reason for not doing so.

[a] Exploring and Unleashing the Power of Large Language Models in Automated Code Translation. FSE’24

[b] Spectra: Enhancing the code translation ability of language models by generating multi-modal specifications

[c] Syzygy: Dual code-test C to (safe) rust translation using llms and dynamic analysis

[d] Rustmap: Towards project-scale c-to-rust migration via program analysis and LLM

**Questions:**

1.	Please clarify the novelty of the proposed approach.

2.	Please justify the selection criteria of evaluation projects and explain why HumanEval is a reasonable choice.

3.	Please justify the test sufficiency of the tests in the studied projects.

4.	Please justify the missing baselines in the current evaluation.

---

### Official Review · Reviewer_iEDT · 2025-11-01

**Soundness:** 1
**Presentation:** 2
**Contribution:** 2
**Rating:** 2
**Confidence:** 5

**Summary:**

The authors propose an agentic pipeline towards transpiling C to safe and idiomatic Rust. The paper proposes using 4 agents each specialized towards a given task in the C to Rust conversion.

- The project manager: does dependency analysis on the C project, creates a dependency graph and proposes modules that can be transpiled
- The tech leader takes the relevant modules and dependency graphs from the project manager and then orders them. It also does heuristic merging to prevent overly fragmented tasks.
- Code monkey: It analyzes the C code from the tech leader, performs retrieval (the paper proposes RAG for this), and then generates Rust code. It proposes multiple candidates and filters relevant candidates.
- Error Resolver: Observes compiler errors (if any) , selects the candidate from the code monkey with least amount of compiler errors and tries to address them in an iterative manner.

**Strengths:**

The technique outperforms baselines mentioned in the paper. Further the authors test their technique on various models (DeepSeek-v3), Got-4o, and Claude-3.5-Sonnet, and show consistent gains. The authors further showcase the validity of passing relevant context, repair iterations, and external knowledge.

**Weaknesses:**

Previous techniques like [SYGYZY (Jain et. al.)](https://arxiv.org/abs/2412.14234) also perform tasks like dependency analysis, and module-based translation. How does Rustify compare against techniques like SYGYZY? Further the paper does not compare or discuss techniques such as [VERT (Yang et. al.)](https://arxiv.org/abs/2404.18852) that also produce Safe Rust.

Further, the following details are missing, and need to be provided to understand the validity of the approach and soundness of the approach:

- "strategies used by the model to solve common types of errors" used by the Code Monkey agent: The number of samples in the 2 broad categories. how are the candidates retrieved?
- Details of the granularity of the code modules (lines of code, functions merged using dependency-based merging and line-based merging) proposed by the Tech Lead agent. How do different levels of granularity affect the final performance?  How is the merging done (do the authors use an LLM to do this?)
- Line 185: "Each agent receives sufficient upstream context to ensure semantic preservation and stability" : what sort of context
- Comparison with open source models that are efficient at coding?
- Limited evaluation data : Recent work [CRUST-bench (Khatry et. al.)](https://arxiv.org/abs/2504.15254) proposes 100 benchmarks that test C to safe Rust transpilation over multiple domains.
- Details on number of completions from each model (from the Code Monkey agent) - the paper mentions multiple candidates are retrieved, in how many cases is are the completions correct?
- Ablations on the repair candidates: The paper assumes that taking the code candidate with minimum number of errors would yeild the best performance, but errors might be correlated. Further, the Rust compiler builds in a stage wise manner, hence type errors supersede borrowing errors, hence it seems that the error selection approach is naive.
- Error analysis on failure cases, where does Rustify fail, what sort of errors are harder to resolve? Reasons for low sucdess on libtree, json.h and binn?
- Comparison with other agentic techniques like SWE agent, Codex, ClaudeCode, AutocodeRover etc. Further, while I understand that closed source language models are being updated rapidly, why are deprecated models like gpt-4o and claude-3.5-sonnet used and not latest models like o3 (released in April,2025), gpt-5(August 7), claude-4-sonnet/claude ( released in May) not used?
- No results on idiomaticity: the authors claim that Rustify improves idiomatic Rust generation, but do not quantify or qualitatively analyze how.
- What test cases are used to measure test pass rate? What are the tests? Are they unit tests or integration tests?

**Questions:**

Please refer to weaknesses

---

### Official Review · Reviewer_Qcky · 2025-11-06

**Soundness:** 3
**Presentation:** 2
**Contribution:** 2
**Rating:** 4
**Confidence:** 4

**Summary:**

The paper presents Rustify, a workflow-guided, multi-agent framework for repository-level C-to-Rust translation. It divides the translation process into four specialized LLM-driven stages, context analysis, task planning, translation, and repair,  each handled by a dedicated agent (Projec tManager, Tech Leader, Code Monkey, Error Resolver). The system maintains an experience base to guide future translations and integrates compiler feedback in a tree-search–based repair loop. Evaluations on several benchmarks and open-source repositories show that Rustify achieves gains over some prior work (C2Rust, MetaGPT and Pan et al. 2024).

**Strengths:**

* The paper provides a coherent decomposition of the translation task into modular agents.

* The experimental results (in terms of unsafe ratio, compilation rate, pass rate and CodeBLEU with respect to a reference translation) show progress over the considered baselines.

* The ablation study in Table 3 isolates the contribution of each module, showing that each of them contributes to the final result.

**Weaknesses:**

* Test coverage is not reported for the considered benchmarks. The number of test cases in Table 1 for some benchmarks seems very low (e.g. urlparser only has one test). Without test coverage, it's hard to know how meaningful the pass rates in Table 2 are. If I/O equivalence to the source code is not maintained, then all the other metrics are not very informative.

* Evaluation of correctness performed by the translation agent (Code Monkey) is limited to compilation. Although compilation feedback is incorporated at the sub-task level, test results are only used once the entire repository has been translated and compiled. Thus, I/O equivalence to the source code is never part of the translation or repair feedback loop, and thus not actually used to guide translation.

* There is no discussion of compilation and execution errors present in the translated benchmarks. For instance, binn is completely failing -- nothing even seems to compile. Some discussion of what's happening there would be interesting.

**Questions:**

* What's the test coverage offered by the tests mentioned in Table 1 for each benchmark?

* Does x under binn's Compilation Rate in Table 2 mean that 0% of the translation could be compiled? If so, do you have any explanation for that behaviour?

* How did you obtain the translated Rust reference code?

Minor:
* What does "Code" for HumanEval mean in Table 1?

---

### Note · Authors · 2026-01-27

I have read and agree with the venue's withdrawal policy on behalf of myself and my co-authors.

---

### Meta-Review · Area_Chair_rVrw · 2026-01-04

**Summary:**

This paper presents Rustify, a workflow-guided, multi-agent framework for repository-level C-to-Rust translation. By decomposing translation into structured, role-specific stages and leveraging compiler feedback and experience reuse, Rustify effectively resolves cross-file dependencies, ensures memory safety, and produces high-quality, idiomatic Rust code. Experiments show that Rustify performs well on
memory-safe Rust code, reducing unsafe code by up to 99% compared to prior approaches, and raises the test pass rate from 10.5% in the baseline to 86.4%.

**Reviewer Concerns:**

The reviewers have several concerns: (1) Unclear novelty: Rustify largely overlaps with prior work (e.g., SYZYGY, AlphaTrans, SWE-Agent) in dependency analysis, module-based translation, multi-agent workflows, and compiler-error feedback, with insufficient justification of what is fundamentally new. (2) Missing and weak baselines: The paper does not compare against several relevant and recent C-to-safe-Rust translation methods, nor discuss why they are excluded. (3) Insufficient method details: Key design choices—such as module granularity, context passed between agents, and error repair strategies—are under-specified, making it hard to assess soundness and reproducibility. (4) Inadequate evaluation: Benchmark selection is questionable, test suites are very small with no coverage analysis, and correctness is mainly measured by compilation success rather than semantic or I/O equivalence. (5) Lack of failure analysis: The paper does not analyze failure cases or explain poor performance on some repositories. Overall, the reviewers find the evaluation and novelty insufficient and request clearer technical explanations and stronger experimental validation.

**Reviewer Scores:**

Unchanged for all reviewers as no rebuttal are provided.

---

### Decision · Program_Chairs · 2026-01-26

Reject